# Changes in Bone Marrow Fatty Acids Early after Ovariectomy-Induced Osteoporosis in Rats and Potential Functions

**DOI:** 10.3390/metabo13010036

**Published:** 2022-12-26

**Authors:** Sizhu Wang, Cuisong Tang, Jieying Chen, Huan Tang, Lin Zhang, Guangyu Tang

**Affiliations:** 1Department of Radiology, Shanghai Tenth People’s Hospital, Tongji University School of Medicine, Shanghai 200072, China; 2Department of Radiology, Clinical Medical College of Shanghai Tenth People’s Hospital of Nanjing Medical University, Shanghai 200072, China; 3Department of Radiology, Huadong Hospital of Fudan University, Shanghai 200040, China

**Keywords:** osteoporosis, lipids, fatty acid, GC-MS

## Abstract

The aim of this study was to investigate the changes in bone marrow fatty acids early after ovariectomy-induced osteoporosis in rats, and explore the potential function of the bone marrow fatty acids. Ninety-six female Sprague Dawley rats (12 weeks) were randomly divided into an ovariectomized (OVX) group and Sham group (N = 48/group) and received ovariectomy or Sham surgery, respectively. After 3, 5, 7,14, 21 and 28 days, eight rats in each group were sacrificed to detect the composition of bone marrow fatty acids by means of gas chromatography-mass spectrometry and evaluate the trabecular bone microarchitecture by means of microCT. Bone marrow rinsing fluid and serum were collected for the detection of nitric oxide synthase/nitric oxide (NOS/NO) and bone metabolism related parameters, respectively. Our results demonstrated that the bone microstructure was damaged significantly from 14 days after OVX surgery onwards. Sample clustering and group separation were observed between the OVX group and Sham group 3 and 14 days after surgery, which suggested the role of bone marrow fatty acids in the early stage of postmenopausal osteoporosis. Palmitoleate, myristate and arachidonate were found to play an important role in classification between the OVX group and Sham group on the 3rd day after surgery (VIP > 1, *p* < 0.05). Palmitoleate, myristate, alpha linolenate, stearate and eicosenoate were found to play an important role in classification between the OVX group and Sham group on the 14th day after surgery (VIP > 1, *p* < 0.05). The levels of myristate, palmitoleate, alpha linolenate and eicosenoate were significantly decreased in the OVX group, while the levels of arachidonate and stearate were significantly increased in OVX group (*p* < 0.05). Additionally, myristate, palmitoleate, alpha linoleate and eicosenoate were negatively correlated with C-terminal telopeptide of type 1 collagen (CTX-1, a bone resorption marker), while arachidonate was negative correlated with osteocalcin (OCN, a bone formation marker) (*p* < 0.05). A significant correlation was also found between eicosenoate and NOS (*p* < 0.05). Profound bone marrow fatty acids changes have taken place in the early stage of post-menopausal osteoporosis. They may affect bone formation though affecting the differentiation and function of osteoclasts or osteoblasts, respectively. The NOS/NO system may mediate the influence of eicosenoate on bone formation.

## 1. Introduction

Osteoporosis (OP) is a common chronic progressive metabolic skeletal disease characterized by low bone mass and degeneration of the bone microarchitecture. OP increases bone fragility and the risk of fractures, causing enormous financial and health consequences [1]. The prevalence of OP is significantly higher in the female population and older age groups [2]. The most common primary OP is postmenopausal osteoporosis (PMOP) [3]. OP is related tomany other metabolic diseases, such as obesity [4]. Additionally, it was found the accumulation of lipid in the bone marrow could be abserved as bone mineral density decreased [5,6]. The composition of lipid in the bone marrow is regarded to have an important influence on or association with bone metabolism and can be linked to OP as marrow lipid is a major component of the bone trabecular microenvironment [7], but the underlying mechanism is not fully understood.

Gas chromatography-mass spectrometry (GC-MS)-based analytical strategies have proven to be powerful tools for detecting fatty acids in biological samples [8]. The identification and quantitation of fatty acids have been applied in various disease to explore possible pathologic mechanism, as well as OP [9,10]. Fatty acids not only in the blood serum, but also in the bone marrow, were analyzed by GC-MS [11,12]. Previous studies suggested an association between increased an unsaturated/saturated index of bone marrow lipid and bone health [12]. Additionally, the specific influence of fatty acids on bone metabolism differs according to different classification. Palmitate, a long-chain saturated fatty acid (LCSFA), could impede the survival and differentiation of osteoblasts and enhance osteoclast differentiation [13]. However, long-chain monounsaturated fatty acids (LCMUFAs) such as oleic acids showed potential against palmitate-induced lipotoxicity to the bone [14,15]. In addition, omega-3 long-chain polyunsaturated fatty acids (LCPUFAs) such as docosahexaenoic acid (DHA) and eicosapentaenoic acid (EPA) could inhibit inflammatory stress to help preventing bone loss while omega-6 LCPUFAs showed the opposite functions [16]. The effect of fatty acids on bone metabolism is complicated [14].

Nitric oxide (NO) is involved in bone remodeling. It appears to have a dual effect on bone formation [17]. NO has become a novel target that has the potential to treat osteoporosis [18,19]. Fatty acids were found to affect NO production. In the studies about endothelial mechanisms, omega-3 PUFAs enhanced NO production partly by means of the up-regulation of endothelial nitric oxide synthase (eNOS) [20]. EPA was found to translocate eNOS to the soluble fraction and increase it activity [21]. EPA and DHA could reduce reactive oxygen species (ROS) and then attenuate the inhibition of NO production by ROS [22]. On the other hand, arachidonic acid could inhibit NOS-I-dependent NO production and was regarded as an important regulation node between the production of physiological and pathological NO [23,24]. Therefore, NO may also mediate the influence of bone marrow fatty acids in osteoporosis [25].

To our knowledge, most fatty acids profiles of OP were mainly obtained in cross-sectional studies. The change in bone marrow fatty acids during the formation of OP remains unknown. Here, we hypothesize that bone marrow fatty acids have changed in the early stage of post-menopausal osteoporosis and contribute to bone loss. Therefore, in our study, to investigate the changes in the environment in the marrow niche and better understand the pathogenesis of OP, we constructed PMOP rat models through ovariectomized (OVX) surgery and analyzed the changes in the fatty acids in the bone marrow at different time points after surgery. In addition, we also explored the relationship between fatty acids and bone microstructure indexes, bone metabolism related parameters and the NOS/NO system, in order to provide a new basis for the potential mechanism of early postmenopausal bone mass loss.

## 2. Results

### 2.1. The Evaluation of Trabecular Bone Microarchitecture after OVX Surgery

The distal femurs were detected by micro-CT at different time points after surgery. The typical micro-CT images in the OVX and Sham groups are shown in Figure 1A.The trabecula of the femurs in OVX group was sparse and the bone cortex became thinner 14 days after surgery compared with the Sham group, which was more obvious after 28 days. The microarchitecture parameters of femurs are shown in Figure 1B. Compared with the Sham group, the values of BMD, BV/TV, Tb.N and Tb.Th were significantly lower in the OVX group, while Tb.Sp was significantly higher in the OVX group from 14 days after surgery onwards. These results show that the OP model was established successfully, and the bone microstructure was damaged significantly from 14 days after OVX surgery onwards.

### 2.2. Differential Bone Marrow Fatty Acids after OVX Surgery

Orthogonal partial least squares-discriminant analysis (OPLS-DA) was used to analyze the clustering feature between the OVX group and the Sham group. An obvious separation trend between two groups was observed on the 3rd day after surgery. The OVX and Sham groups were relatively discrete on the 14th day (Figure 2A,D). The permutation test showed that R2Ys were 0.832 and 0.737, and Q2s were 0.492 and 0.48, respectively, on the 3rd day and 14th day after surgery (Figure 2B,E, *p* < 0.05). The result indicates that the established OPLS-DA models had good adaptability and predictability. The differential fatty acids were screened through OPLS-DA by the condition of a variable importance projection (VIP) value over 1 and significant results of the *t*-test. OPLS-DA showed that the VIP values of palmitoleate, myristate and arachidonate were over 1 on the 3rd day. The VIP values of palmitoleate, myristate, alpha linolenate, gamma linolenate, pentadecanoate, stearate and eicosenoate were over 1 on the 14th day (Figure 2C,F). The specific VIP values are shown in Appendix A. The variation trend of these fatty acids is shown in Figure 3. On the 3rd day after surgery, the level of arachidonate was significantly increased in the OVX group compared with the Sham group (*p* < 0.05), while the levels of myristate and palmitoleate were both decreased in the OVX group compared with the Sham group (*p* < 0.05). The levels of pentadecanoate, stearate, alpha linoleate, gamma linoleate and eicosenoate were not significantly changed (*p* > 0.05). On the 14th day after surgery, the level of stearate was significantly increased in the OVX group compared with the Sham group (*p* < 0.05), while the levels of alpha linoleate, eicosenoate, myristate and palmitoleate were significantly decreased in the OVX group(*p* < 0.05). The levels of arachidonate, pentadecanoate and gamma linoleate were not significantly changed (*p* > 0.05). Additionally, there was no obvious separation trend between the OVX and Sham groups on the 5th, 7th, 21st and 28th days (Figure 4). Therefore, according to the result of the OPLS-DA and *t*-test, the differential fatty acids on the 3rd day were palmitoleate, myristate and arachidonate, and the differential fatty acids on the 14th day were palmitoleate, myristate, alpha linolenate, stearate and eicosenoate (Table 1).

### 2.3. Bone Marrow Fatty Acids Levels Were Correlated with Trabecular Bone Microarchitecture

We performed correlation analysis of the OVX group and Sham group together and the relationships between differential bone marrow fatty acids and trabecular bone microarchitecture are showed in Figure 5. On the 3rd day (Figure 5A), there was no significant correlation between myristate, palmitoleate, arachidonate and trabecular bone microarchitecture (*p* > 0.05). However, there was an obvious negative correlation coefficient between arachidonate and BV/TV (r = −0.49) and an obvious positive correlation coefficient between arachidonate and Tb.Sp (r = 0.60), though the *p* value was not statistically significant. On the 14th day (Figure 5B), myristate was positively correlated with BV/TV (r = 0.78, *p* < 0.05). Palmitoleate was positively correlated with BMD (r = 0.83, *p* < 0.05), while stearate was negatively correlated with BMD (r = −0.86, *p* < 0.05). Although there was no significant correlation between alpha linolenate, eicosenoate and trabecular bone microarchitecture (*p* > 0.05), obvious correlation coefficients were still found. There was a positive correlation coefficient between alpha linolenate and BMD (r = 0.58) and BV.TV (r = 0.62). There was a positive correlation coefficient between eicosenoate and BV.TV (r = 0.64) and Tb.N (r = 0.61).

### 2.4. Bone Marrow Fatty Acids Levels Were Correlated with Bone Metabolism-Related Parameters and NOS

To explore how the bone marrow fatty acids affect bone metabolism, we further investigated the correlation between significant bone marrow fatty acids and osteoblastic and osteoclastic parameters osteocalcin (OCN), N-terminal propeptide of type I procollagen (PINP) and C-terminal telopeptide of type 1 collagen (CTX-1), as well as the NOS/NO system, in the OVX group and Sham group together. The results are shown in Figure 6. On the 3rd day after surgery (Figure 6A), palmitoleate was found to be negatively correlated with CTX-1 (r = −0.51, *p* < 0.05), while arachidonate was negatively correlated with OCN (r = −0.76, *p* < 0.05). No significant correlation was found between these bone marrow fatty acids and PINP, as well as the NOS/NO system (*p* > 0.05). On the 14th day after surgery (Figure 6B), myristate (r = −0.67, *p* < 0.05), alpha linolenate (r = −0.56, *p* < 0.05) and eicosenoate (r = −0.65, *p* < 0.05) were negatively correlated with CTX-1. Additionally, eicosenoate was positively correlated with NOS (r = 0.74, *p* < 0.05). No significant correlation was found between these bone marrow fatty acids and osteoblastic parameters (PINP and OCN) and NO. The significant results of correlation analysis are summarized in Table 2.

## 3. Discussion

Accumulating evidence has suggested the important roles of bone marrow fatty acids in bone metabolism [14]. However, due to the special structure of the skeletal niche, functional research on bone marrow fatty acids in situ has been difficult. The role of bone marrow fatty acids still needs further study. In our study, we investigated the changes in bone marrow fatty acids early after ovariectomy-induced osteoporosis in rats and explored the possible mechanism of bone marrow fatty acids affecting bone metabolism by means of correlation analysis in the early stage of osteoporosis formation.

Estrogen deficiency leads to changes in bone marrow fatty acids. An apparent sample clustering and group separation was observed between OVX group and Sham group on the 3rd and 14th days after surgery, suggesting the role of bone marrow fatty acids in the early stage of PMOP. VIP of OPLS-DA is considered as an effective method to select useful substance for classification [26]. According to the results of the VIP value and *t*-test, arachidonate played an important role in classification between OVX group and Sham group on the 3rd day after surgery. Stearate, alpha linolenate and eicosenoate were significant for classification on the 14th day after surgery. Myristate and palmitoleate were both significant on the 3rd and 14th day after surgery.

Stearate belongs to the LCSFAs, which are generally regarded to enhance bone resorption and inhibit bone formation [27]. In our study, we found that the level of stearate was increased in the OVX group and negatively correlated with BMD on the 14th day after surgery, which indicated that the increased level of stearate may attenuate the bone formation. Myristate is also a kind of LCSFA, but its level was significantly decreased in the OVX group on both the 3rd and 14th day after surgery. In addition, myristate was positively correlated with BMD and BV/TV on the 14th day after surgery, which suggested its positive role in bone formation. Previous studies indicated that myristoleic acid could inhibit osteoclast formation and bone resorption [28], and we also found that there was a negative correlation between myristate and CTX-1, which is a serum biomarker of osteoclast related bone resorption. These results suggest that the decreased level of myristate in the OVX group may attenuate the inhibition of osteoclastogenesis on the 3rd and 14th days after surgery.

The effects of unsaturated fatty acids on bone metabolism are more complicated. According to the number of unsaturated bonds, long-chain unsaturated fatty acids were classified into LCMUFAs such as omega-5, omega-7 and omega-9 and LCPUFAs such as omega-3 and omega-6.

Alpha linolenate is a kind of omega-3 LCPUFAs, which could protect against inflammation and contribute to the maintenance of bone mineral density [29,30]. The level of alpha linoleate was significantly decreased in the OVX group and negatively correlated with CTX-1 on the 14th day after surgery in our study, which is consistent with previous studies showing that alpha linoleate could inhibit osteoclastogenesis and prevent inflammatory bone loss [31]. Arachidonate (omega-6 LCPUFAs), considered to be pro-inflammatory and to increase bone loss [16,32], was found to be positively correlated with Tb.Sp on the 3rd day after surgery in our study. We also found that arachidonate was negatively correlated with OCN. Previous studies have demonstrated that arachidonate could inhibit differentiation and osteoprotegerin (OPG) secretion of MC3T3-E1 osteoblast-like cells [33,34]. These results suggest that the decreased level of alpha linoleate may function by attenuating the inhibition of osteoclast differentiation while the increased level of arachidonate may function by inhibiting the function of osteoblasts in the early stage of osteoporosis.

LCMUFAs are generally believed to protect bone health [14]. The level of palmitoleate (omega-7 LCMUFAs) was decreased in the OVX group on the 3rd and 14th days after surgery, and the level of eicosenoate (omega-9 LCMUFAs) was decreased in the OVX group on the 14th day after surgery. Additionally, palmitoleate was found to be positively correlated with BMD on the 14th day. Previous studies have demonstrated that palmitoleate could inhibit osteoclastic differentiation and promote osteoblast signaling [35,36]. We also found a negative correlation between palmitoleate and CTX-1 on the 3rd day after surgery, which suggested that a lower level of palmitoleate in the OVX group may decrease the suppression of osteoclastogenesis in the early stage of osteoporosis. In addition, there was a negative correlation between eicosenoate and CTX-1 on the 14th day after surgery as well, which also indicated less suppression on osteoclastogenesis. Interestingly, to our knowledge, there is little research indicating the function of eicosenoate on bone metabolism, and these findings need further investigation.

The function of nitric oxide (NO) in preventing bone loss has long been investigated. NO is synthesized from arginine by nitric oxide synthases (NOS) [37]. NO is considered to promote the differentiation and proliferation of osteoblasts, because it can promote aerobic glycolysis, which is the energy source of osteoblasts [38]. However, the effects of NO on osteoclasts are complicated. It has been suggested that NO generated by inducible NOS (iNOS) can inhibit osteoclastogenesis while NO generated by constitutive NOS (cNOS) promotes this process [39]. Additionally, exogenous NO can prevent the function of osteoclasts [40]. In our study, we found that eicosenoate was positively correlated with NOS, but no significant correlation was found between these fatty acids and NO. This indicates that the NOS/NO system might mediate the influence of eicosenoate on bone metabolism. Some studies on other diseases have discovered that fatty acids function by affecting the NOS/NO system. For example, the concentrations of oleic and palmitic acids could affect the oxidative status caused by reactive oxygen species in vascular endothelium [41]. Dietary omega-3 LCPUFA supplementation improved platelet NOS function in type 2 diabetes mellitus [42]. Since we only measured the total content of NOS and NO in the cells from bone marrow, and we did not detect iNOS and cNOS separately, more studies are needed to explore the effect of eicosenoate on the NOS/NO system in osteoblasts and osteoclasts, respectively.

Taken together, we inferred that bone marrow fatty acids significantly change in the early stage of osteoporosis and affect bone metabolism. Some bone marrow fatty acids were correlated with CTX-1 or OCN, indicating that these fatty acids may function by affecting the differentiation and function of osteoclasts or osteoblasts, respectively. There was also a significantly positive correlation between eicosenoate and NOS, which suggested that the NOS/NO system may also be involved in the influence of fatty acids. These findings demonstrated that the function of bone marrow fatty acids is complicated because we did not find any specific bone marrow fatty acids that were significant at each studied time point after surgery. The ratio and metabolism between fatty acids also affected the bone metabolism while we did not discuss in this study [12,43,44]. Therefore, further investigations are required to confirm the correlation discovered in this study.

There are some limitations to this study. Firstly, 3-month-old rats were used for the experiment, but the age of the rats may be relatively young. The bones of the rats may still be growing due to the influence of growth hormones, so the situation of postmenopausal osteoporosis in women may not be completely simulated. Secondly, the trabecular bone microarchitecture was obtained at the femur, while the bone marrow fatty acids were obtained from the femur and humerus. The difference between bones may affect the results. Thirdly, we did not measure the levels of NOS and NO, respectively, in osteoblasts and osteoclasts. The levels of CTX-1, PINP and OCN were measured in the serum. The correlations and speculated potential mechanism need further specific experiments to demonstrate. Finally, we did not discuss the interaction and the ratio between fatty acids, which also contribute to the bone metabolism.

## 4. Materials and Methods

### 4.1. Animals and Establishment of Osteoporotic Rat Model

The Animal Ethics Committee of Shanghai Tenth People’s Hospital approved the experimental protocol and animals feeding method in this experiment. A total of 96 female Sprague-Dawley (3-month-old) rats were purchased from Vital River Laboratory Animal Technology Co., Ltd. Animals were fed freely and accommodated in a well-ventilated clean animal room with a 12/12-h light/dark cycle. Rats were randomly classified into the sham operated group (Sham) and ovariectomized (OVX) group, with 48 rats in each group. After rats were anaesthetized with isoflurane (RWD life science). Bilateral ovariectomy was performed in the OVX group. The oviduct was ligated carefully. In the Sham group, rats underwent a sham surgery, and the same mass of adipose tissue around the ovary was removed.

### 4.2. Specimen Collection

On the 3rd, 5th, 7th, 14th, 21st and 28th days after the operation, eight rats in each time point of the two groups were anesthetized and then 5-mL of blood was collected by heart puncture. The blood was centrifuged for 15 min at 650× *g*, and the serum was stored at −80 °C for subsequent experiments. The femurs and humerus were taken, and the muscles and soft tissues were removed from the bones. The right femur and humerus samples were flushed with phosphate buffer saline (PBS), and then placed in a −80 °C refrigerator for Gas chromatography/mass spectrometry (GC-MS) examination. Bone marrow cells were flushed from the left humerus and then placed in the −80 °C refrigerator for detection of NOS and NO. The left femurs were fixed in a paraformaldehyde solution for micro-computerized tomography (micro-CT) examination.

### 4.3. Bone Microstructure Measurement

The distal femur of rats was assessed with skyscan1176 μCT (Bruker micro CT, Kontich, Belgium). NR econ (Bruker micro CT, Belgium) software was used for tomography reconstruction. Based on the tomography images, CT an (Bruker micro CT, Belgium) software was used to analyze the bone microstructure parameters of trabecular bone and the pictures with a voxel size of 17.93 μm were taken at an energy of 90 kV and intensity of 278 μA. The measurement parameters included: BMD, BV/TV, Tb.N, Tb.Th and Tb.Sp.

### 4.4. Gas Chromatography/Mass Spectrometry

Bone PBS rinse liquid was put into a high-throughput tissue lapping apparatus with 1 mL of a chloroform: methanol (2:1) solution and 100 mg glass beads, and shaken at 60 Hz for 1 min, repeated twice. After that, 1% sulfuric acid methanol solution was used for esterification and hexane was used for extraction. One hundred milligrams of anhydrous sodium sulfate powder was added into the supernatant to remove excess water, and the supernatant was subjected to gas chromatography/mass spectrometry analysis. The chemical formulas of fatty acids are listed in Appendix A.

### 4.5. Determination of Serum CTX-1, OCN and PINP Levels

Enzyme-linked immunosorbent assay kits (Shanghai Lengton Bioscience Co., Ltd., Shanghai, China) were used to measure the serum levels of CTX-1, OCN and PINP, according to the manufacturer’s instructions.

### 4.6. NO Measurement and NOS Enzymatic Activity Assay

Bone marrow cells was lyzed with Cell and Tissue Lysis Buffer for the Nitric Oxide Assay (Beyotime, Shanghai, China), and the measurement of NO content was carried out with the NO assay kit (Beyotime, Shanghai, China), according to the manufacturer’s instructions. NOS enzymatic activity in bone marrow cells was measured according to the manufacturer’s instructions with the Nitric Oxide Synthase Assay Kit (Beyotime, Shanghai, China). A fluorescence microplate reader was used to measure fluorescence at an excitation of 495 nm and an emission of 515 nm.

### 4.7. Statistical Analysis

MetaboAnalyst version 5.0 (https://www.metaboanalyst.ca/, accessed on 10 August 2022) [45] was used to process the fatty acids data. A multivariate analysis (OPLS-DA) was carried out to identify separation or clustering between the OVX group and the Sham group. Statistically significant fatty acids that meet the VIP values greater than 1 were deemed as potential fatty acids. Data were presented as the mean ± standard error of the mean (SEM). Comparisons between two groups were performed using Student’s unpaired *t*-test. Correlations between fatty acids and the level of the NOS/NO system and serum parameters were evaluated with a Spearman rank correlation test by GraphPad prism software (GraphPad Software, USA, accessed on 30 January 2019). A value of * *p* < 0.05, ** *p* < 0.01, and *** *p* < 0.001 was considered statistically significant.

## 5. Conclusions

This study identified several bone marrow fatty acids, which were significant changed in the early stage of PMOP. These include myristate, palmitoleate, arachidonate, stearate, alpha linolenate and eicosenoate. In addition, the correlation analysis indicated that these significant bone marrow fatty acids may affect the bone microarchitecture. The findings of this study could add a layer of information toward understanding the possible bone metabolism alterations in the early stage of PMOP.

## Figures and Tables

**Figure 1 metabolites-13-00036-f001:**
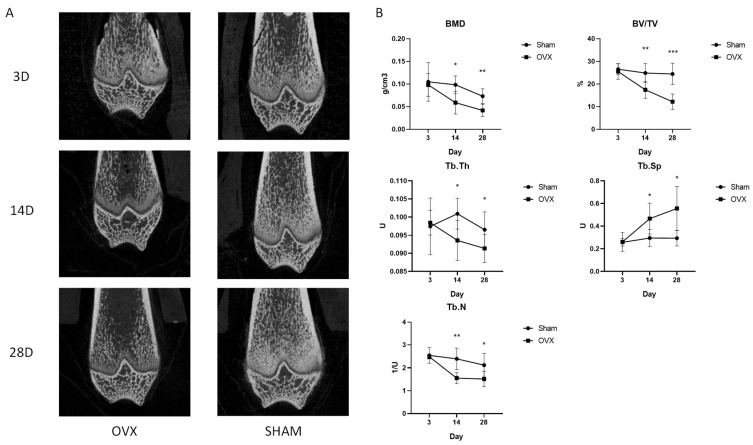
Construction of osteoporosis model. (**A**) The typical microCT 2D images of femurs. (**B**) The microarchitecture parameters of bone in the distal femurs of rats after OVX surgery, including bone mineral density (BMD, unit: g/cm^3^), bone volume fraction (BV/TV, unit: %), trabecular thickness (Tb.Th, unit: U), trabecular spacing (Tb.Sp, unit: U), and trabecular number (Tb,N, unit: 1/U). * *p* < 0.05, ** *p* < 0.01, *** *p* < 0.001.

**Figure 2 metabolites-13-00036-f002:**
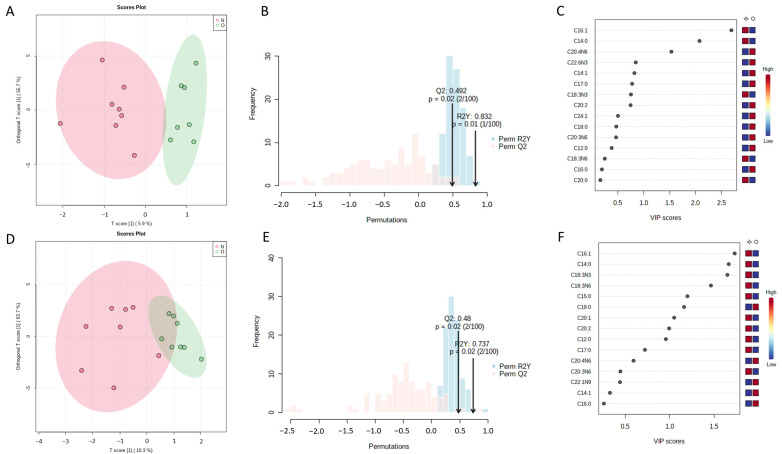
Multivariate statistical analysis of GC-MS data. OPLS-DA score plots and permutation test results of bone marrow fatty acids 3 (**A**,**B**) and 14 (**D**,**E**) days after OVX surgery (*n* = 8 rats per group). (**C**,**F**) VIP plot of the OPLS-DA model. Screening criteria for fatty acids: VIP ≥ 1. N: Sham group, O: OVX group. Color of square indicates fatty acids content, red: high content and blue: low content.

**Figure 3 metabolites-13-00036-f003:**
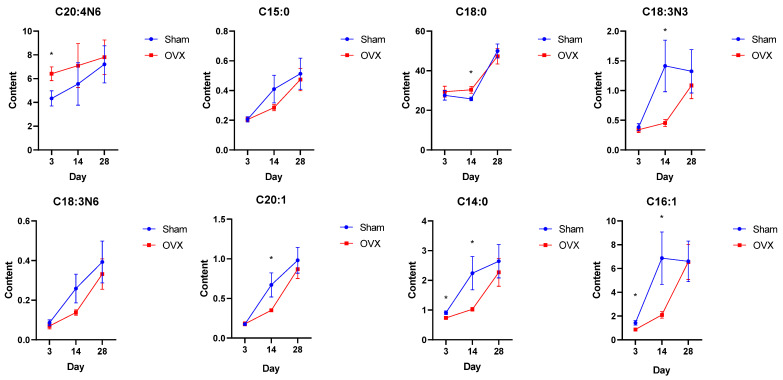
Concentration changes for significant bone marrow fatty acids after surgery. * *p* < 0.05.

**Figure 4 metabolites-13-00036-f004:**
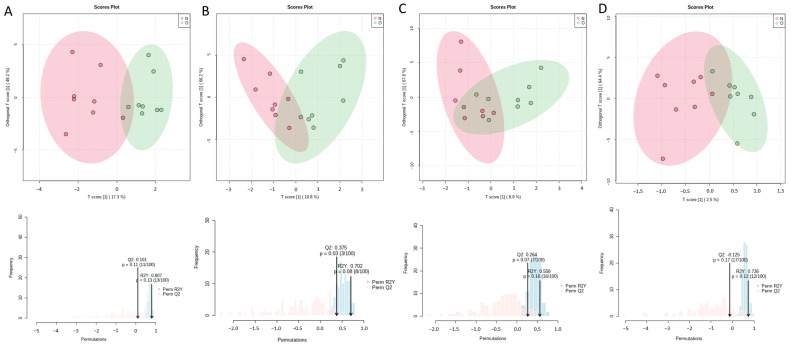
Multivariate statistical analysis of GC-MS data. OPLS-DA score plots and permutation test results of bone marrow fatty acids 5 (**A**), 7 (**B**), 21 (**C**) and 28 (**D**) days after OVX surgery (*n* = 8 rats per group). N: Sham group, O: OVX group.

**Figure 5 metabolites-13-00036-f005:**
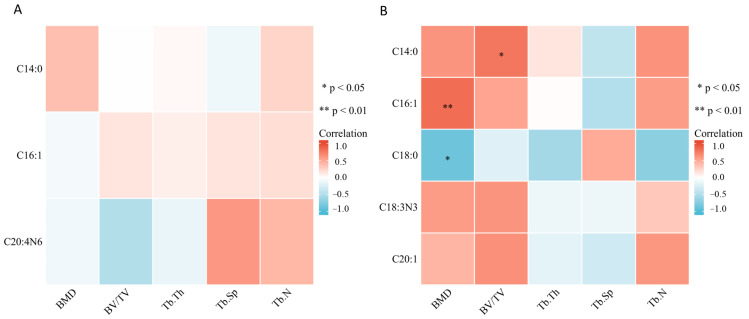
Correlation between differential fatty acids and bone microarchitecture parameters. (**A**) Correlation between differential fatty acids and bone microarchitecture parameters on the 3rd day after surgery. (**B**) Correlation between differential fatty acids and bone microarchitecture parameters on the 14th day after surgery. * *p* < 0.05 and ** *p* < 0.01 suggest a statistically significant correlation. Color key indicates the correlation coefficients, red: positive correlation, blue, negative correlation.

**Figure 6 metabolites-13-00036-f006:**
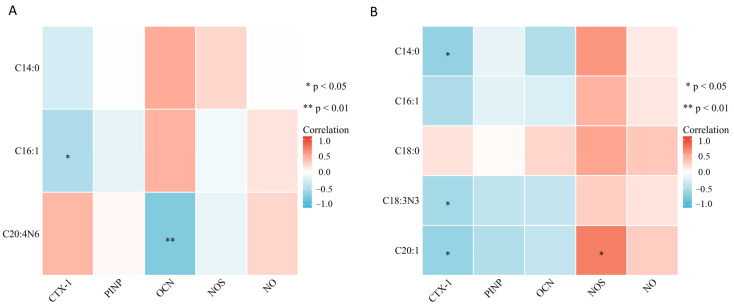
Correlation between differential fatty acids and serum bone metabolism-related parameters and the NOS/NO system. (**A**) Correlation between differential fatty acids and serum bone metabolism-related parametersand the NOS/NO system on the 3rd day after surgery. (**B**) Correlation between differential fatty acids and bone metabolism-related parameters and the NOS/NO system on the 14th day after surgery. * *p* < 0.05 and ** *p* < 0.01 suggest a statistically significant correlation. Color key indicates the correlation coefficients, red, positive correlation, blue, negative correlation.

**Table 1 metabolites-13-00036-t001:** OPLS-DA and *t*-test results of fatty acids.

Day	Fatty Acid	OVX Group (Mean ± SEM)	Sham Group (Mean ± SEM)	*p* Value	VIP Value
3rd day	Palmitoleate	0.8749 ± 0.1389	1.430 ± 0.1916	0.040	2.69
	Myristate	0.7399 ± 0.03411	0.9102 ± 0.06195	0.033	2.08
	Arachidonate	6.414 ± 0.5745	4.341 ± 0.6434	0.037	1.53
14th day	Palmitoleate	2.099 ± 0.2807	6.868 ± 2.200	0.038	1.73
	Myristate	1.028 ± 0.0602	2.241 ± 0.5614	0.038	1.66
	Alpha linolenate	0.4549 ± 0.0600	1.415 ± 0.4345	0.036	1.65
	Stearate	30.38 ± 1.737	25.76 ± 0.8923	0.046	1.16
	Eicosenoate	0.3501 ± 0.01103	0.6707 ± 0.1521	0.045	1.05

**Table 2 metabolites-13-00036-t002:** Correlation coefficients of significant differential fatty acids with bone microstructure and bone metabolism related parameters.

Time	Fatty Acid	Bone Microstructure			Bone Metabolism Related Parameters		
		BMD	BV/TV	Tb.Sp	CTX-1	OCN	NOS
3rd day	Myristate						
	Palmitoleate				−0.51		
	Arachidonate			0.6		−0.76	
14th day	Myristate	0.62	0.78		−0.67		
	Palmitoleate	0.83					
	Stearate	−0.86					
	Alpha linolenate				−0.56		
	Eicosenoate				−0.65		0.74

Above results are all statistically significant (*p* < 0.05).

## Data Availability

The data presented in this study are available on request from the corresponding author. The data are not publicly available due to privacy.

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
