# Peer review of "Changes in Bone Marrow Fatty Acids Early after Ovariectomy-Induced Osteoporosis in Rats and Potential Functions"

_metabolites, 2022, doi:10.3390/metabo13010036_

Round 1

Reviewer 1 Report

In the presented article, "Changes of Bone Marrow Early Stage after Ovariectomy-Induced Osteoporosis in Rats and Potential Functions"

The article is well-structured and well-written, and I highly recommend this article for publication in Metabolites Journal. 

Minor point.

-       The quality of the figures (2, 4, and 5) is blurred. Please improve them. 

-       (Line 150) I did not see separation; I see overlap between groups after 14 days

-       (Line 105) put punctuation colon (:) between chloroform and methanol to become chloroform: methanol

-       (Line 63) needs a reference.

-       (Line 124) needs a reference on MetaboAnalyst

-       (Line 125) OPLS-DA, you should write the full name at the first mention and put the abbreviation in brackets.

-       Although the manuscript reads well overall, there is a minor grammar error that needs to be corrected.

Reviewer 2 Report

Comments on Metabolites

The manuscript by Shizhu Wang et al identified bone marrow fat acids change during ovariectomy induced osteoporosis development. The study used gas chromatography-mass spectrometry as main method to identify many different fat acids change earlier after ovariectomy and performed correlation analysis between the level of these fat acids with bone microarchitecture parameters as well as NOS/NO and bone metabolism markers. The study revealed very important information for future studies of these fat acids and osteoporosis development. The study is novel and represents large amount of work and fit the Journal topic. However, several improvements are needed to make this manuscript acceptable for publication in Metabolites.

Following are the detail comments:

Figure 1,2,3,4,5 fonts are too small and fonts are in gray color, it is hard to read. Please enlarge in original GraphPad Software or type in Photoshop to make the figure more appealing and clearer. Figure 3 has too much space between upper and lower panel. Make it more compact and make chart bigger.

Add more descriptions on figure legends.

English grammars need to be improved in general.  Please read carefully to express your meaning accurately.

Since the manuscript only have 5 figures and 2 tables, I suggest bring supplemental data to main figures to increase the readability of the manuscript.

Abstract: Line 19:”As results”, grammar issue, may be “Our results demonstrated”. 

Results:

Line 143,  “OP model was constructed successfully”. Here the “constructed” does not correctly express the meaning well, maybe “established”

Did the authors perform correlation analysis of OVX group and Sham group together for each time points? Or only OVX group mice? Please clarify.

Discussion:

Line 224, “The special construction of skeletal niche” The word “construction”. better be replaced with “structure”.

Line 256, “contribute to the bone mineral density” meaning is not clear to me, are the authors intend to say “contribute to the maintenance of bone mineral density?

Conclusion section:

1.       The authors should include correlations of fat acid with bone microarchitecture parameters while they cannot conclude anything on correlation with osteoblasts or osteoclast since they did not performed staining of the bones to identify osteoblasts and osteoclasts number.

2.       Line 319, “This study identified several bone marrow fatty acids, which were significant in the early stage of PMOP” Meaning is not clear. Did the author mean” Which were significantly changed?

3.       Line 320, “There were” Should be “These include”.

Reviewer 3 Report

The study assessed the effect of ovariectomy on fatty acid composition of bone marrow and how this might influence bone formation and resorption. The study findings are of interest. The manuscript however requires substantial English editing.

Abstract, line 19: Define the abbreviation “NOS/NO”

Line 28: “Besides” is the wrong word to use here. I suggest replacing with “Also”

Line 29: Again, please spell out the abbreviations used on this line: CTX-1, OCN. I suggest also indicating that one is a marker of resorption and one a marker of bone formation (for the reader who may be unfamiliar with these markers).

Line 43: Replace “Besides” with “Also”. Replace “it suggested” with “it is suggested”. I suggest re-wording this entire sentence, as it sounds awkward.

Line 55: Replace “Besides” with “Also”

Line 56: “Palmitate, a kind of long-chain saturated fatty acids”…Reword to “Palmitate, a long-chain saturated fatty acid…” Also, a reference is needed at the end of this sentence where you have indicated this fatty acid affects bone cells.

In the introduction, it would be ideal to indicate how different fatty acids might be involved in nitric oxide synthesis.

The end of the introduction would benefit from one or two hypothesis statements.

Line 75: Replace “ratify” with “approved”

Lines 79-80: “Rats were indiscriminately classified into two groups” It is unclear what was done here. Were the rats randomized to the two groups?

The rats were ovariectomized at 3 months. This is a relatively young age and the rats are still probably growing (this would not simulate the postmenopausal state in women). Please comment on this limitation.

Overall the English in the manuscript needs much improvement. I suggest recruiting an English-speaking colleague or hiring an English-language service to edit the manuscript.

Line 87: “softly anesthetized” – it is unclear what is meant by “softly” here.

Line 96: bone microstructure: This measurement is not mentioned in the abstract. I suggest adding some information on this measurement and results to the abstract if possible.

I think Table 1 should be move to the results section. In the first column of this table, Palmitoleate is listed twice. The first time it is listed, it is unclear which day this corresponds to. I have the same comment for Myristate. Results for this fatty acid appear to be presented twice for the day 3 value.

Line 114: Define the abbreviations “CTX-1, OCN and PINP”

Statistics: Rather than using unpaired students t-tests to evaluate differences between groups at each time point, I would suggest using an ANOVA (or perhaps a MANOVA) to compare the groups across multiple time points, with appropriate post-hoc tests, to protect against type I error. It appears from the legend in Figure 2 you may have used a multivariate statistical test, but this is not described in the statistics section.

For figure 1b, explain all the abbreviations used in the graph titles and axes in the legend to the figure.

Line 168: Change “besides” to “Also”

Line 209: Change “Besides” to “Also”

Line 226: Change “researches” to “study”

Line 239: “LCSFAs” – please ensure this abbreviation is defined in the manuscript

Lines 253 and 263: same comment

Line 271: Change “Besides” to “Also”

Line 281: Change “Then” to “The”

Lines 283-287: References are needed to support statements in these sentences.

Line 287: Change “Besides” to “Also”

Lines 291 and 296: Change “researches” to “studies”

Please see the following study, which might be related to your study. This could be included in your introduction or discussion section:

https://pubmed.ncbi.nlm.nih.gov/31796982/

Reviewer 4 Report

The authors investigate the changes of environment in the  marrow niche and better understand the pathogenesis of OP, we constructed PMOP rat  models through OVX surgery and analyzed the changes of fatty acids in bone marrow at  different time points after surgery. In addition, we also explored the relationship between fatty acids and bone microstructure indexes, bone metabolism related parameters and  NOS/NO system, in order to provide a new basis for potential mechanism of early post-71 menopausal bone mass loss. This study identified several bone marrow fatty acids, which were significant in the  early stage of PMOP. There were myristate, palmitoleate, arachidonate, stearate, alpha-linolenate and eicosenoate. In addition, correlation analysis indicated that these significant bone marrow fatty acids may function by affecting osteoblasts or osteoclasts. NOS/NO system may mediate the influence of eicosenoate on bone formation. The findings of this study could add a layer of information to understanding the possible bone  metabolism alterations happens in the early stage of PMOP.

The introduction is well written , with adequate bibliographic references and stating the hypothesis of the study 

The methodology is complete, widely described, which would allow the study to be carried out by another research group. However, the table 1 must be included in the results section Results are clearly described. The graphic representation  help to understand the results The discussion is correct, adapting to the results obtained. The strengths must be included

Round 2

Reviewer 3 Report

The authors have addressed most of my concerns. As mentioned previously, it would be ideal to have the manuscript edited to improve the English.